# annDNA: Learning Annotation-Aware Genomic Representations via Knowledge Distillation

## Abstract

Genomic language models (GLMs) learn contextual representations of DNA sequences, but current approaches rely solely on sequence patterns without incorporating known genomic and functional annotations. To address this, we present annDNA, which involves two stages: (1) annotation-aware pre-training that creates tokens explicitly encoding functional information from GENCODE and ENCODE, and (2) cross-modal knowledge distillation that transfers these annotation-aware representations to a sequence-only model. Annotation-aware models achieve 15.5% higher AUROC than sequence-only baselines in variant effect prediction. The distilled model, with one-third of the parameters, achieves an 11.2% improvement over the sequence-only baseline while requiring only sequence input at inference. Our results demonstrate the effectiveness of using annotations during training, offering a general framework for transferring biological knowledge to sequence-only models.

## 1. Introduction

Large language models have demonstrated remarkable success in learning contextual representations from natural language. Genomic language models (GLMs) extend this paradigm to DNA sequences, enabling applications such as variant effect prediction, regulatory element classification, and gene expression modeling (Benegas et al., 2025c). Recent advances have further expanded GLM capabilities, including functional element discovery through context analysis (Tomaz da Silva et al., 2025) and *de novo* gene generation (Merchant et al., 2026).

Despite recent progress, existing GLMs encode DNA using

[1]Anonymous Institution, Anonymous City, Anonymous Region, Anonymous Country. Correspondence to: Anonymous Author <anon.email@domain.com>.

Preliminary work. Under review by the International Conference on Machine Learning (ICML). Do not distribute.

sequence information alone, without incorporating known functional annotations. Unlike natural language, which has explicit semantic units such as words and sentences, DNA sequences lack clear boundaries delineating functional elements. As a result, GLMs must infer genomic organization—such as promoters versus enhancers or coding regions versus introns—implicitly from sequence patterns. Various tokenization strategies have been proposed (Ji et al., 2021; Dalla-Torre et al., 2025; Sanabria et al., 2024; Nguyen et al., 2023; Schiff et al., 2024), but all rely solely on sequence context.

This reliance on implicit pattern discovery can be data-inefficient and may fail to capture sharp functional boundaries without large-scale pre-training. In contrast, decades of large-scale experimental efforts have produced comprehensive, experimentally validated annotations of genomic elements (Moore et al., 2020). These annotations provide explicit functional context that is not directly observable from sequence alone, suggesting an opportunity to incorporate external biological knowledge into the training of GLMs.

We present annDNA (annotation-aware DNA), a framework that integrates genomic annotations into input representations through two stages:

- **Annotation-aware pre-training**: We train three BERT-based models with increasing level of annotation complexity: annDNA-seq using sequence only, annDNA-struct adding structural annotations from GENCODE (Mudge et al., 2025), and annDNA-full further incorporating regulatory elements from EN-CODE (Moore et al., 2020).

- **Cross-modal knowledge distillation**: We transfer annotation-aware representations to a sequence-only model via hidden state matching (Sanh et al., 2019), enabling annotation-free inference while retaining most performance gains.

As shown in Figure 1c, annotation-aware models outperform the sequence-only baseline despite the identical architecture. We evaluate these models through genomic region embedding analysis, variant effect prediction benchmarks, and

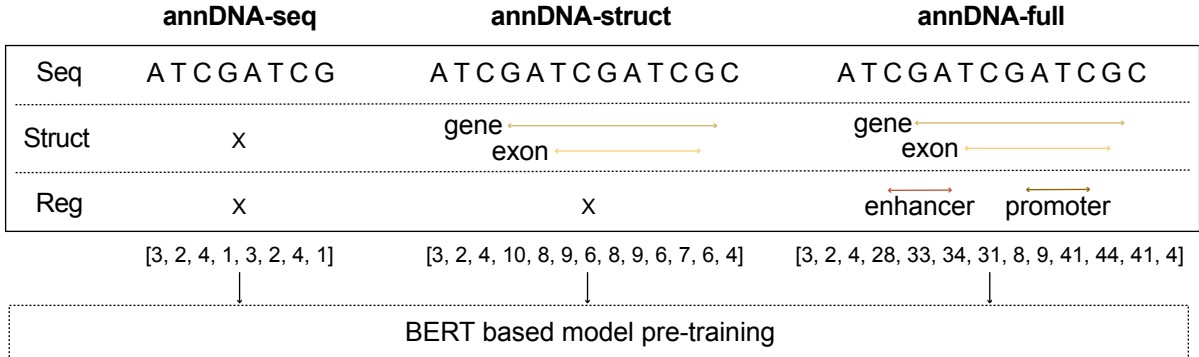

(a) **Stage 1: Annotation-aware pre-training**

(b) **Stage 2: Cross-modal knowledge distillation**

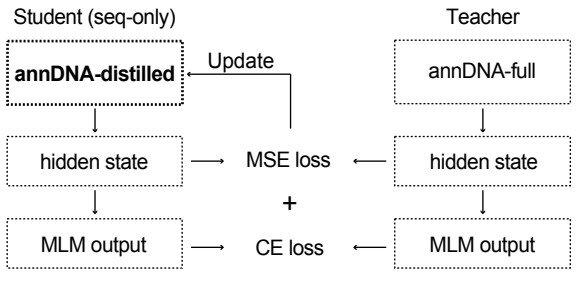

(c) **Overall Performance**

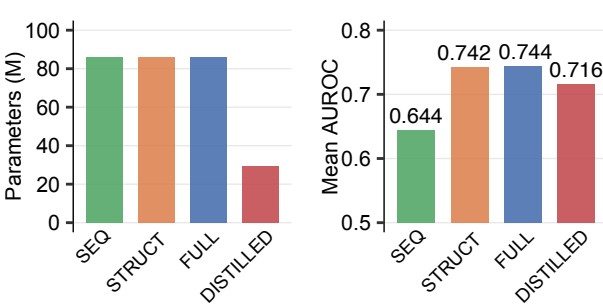

*Figure 1.* **The annDNA framework.** (a) Annotation-aware tokenization for annDNA-seq (sequence-only), annDNA-struct (sequence + structural), and annDNA-full (sequence + structural + regulatory). (b) Cross-modal knowledge distillation from annDNA-full teacher to annDNA-distilled. (c) Annotation-aware models substantially outperform the sequence-only baseline in variant effect prediction, while annDNA-distilled achieves competitive performance with only one-third of the parameters.

attention pattern analysis.

Our contributions are as follows:

- We introduce annDNA, a GLM framework incorporating genomic annotations at an input level, creating tokens that explicitly encode functional information while maintaining single-nucleotide resolution.

- We provide systematic evaluation showing improvements in embedding structure, variant effect prediction, and attention alignment with known functional elements.

- We demonstrate cross-modal knowledge distillation from an annotation-aware teacher to a sequence-only student, enabling practical deployment without annotation requirements.

## 2. Related Work

### 2.1. Genomic Language Models

**Tokenization Strategies.** Tokenization fundamentally shapes how GLMs process DNA sequences. Early approaches employed k-mer tokenization: DNABERT uses overlapping 6-mers to capture local sequence patterns (Ji et al., 2021), while Nucleotide Transformer scales model size and training data to improve performance (Dalla-Torre et al., 2025). However, k-mer tokenization breaks biological units arbitrarily and loses single-nucleotide resolution critical for variant analysis.

Alternative strategies have emerged to address these limitations. GROVER applies byte-pair encoding (BPE) learned from genomic sequences, creating a data-driven vocabulary that captures recurring motifs (Sanabria et al., 2024). Yet recent studies have shown that tokenizer choice induces task-specific trade-offs, with no single method optimal across all downstream tasks (Lindsey et al., 2025). HyenaDNA and Caduceus adopt single-nucleotide tokenization to pre-

serve nucleotide-level resolution, enabling long-range context modeling through architectural innovations (Nguyen et al., 2023; Schiff et al., 2024). Recent work explores learnable tokenization: MxDNA employs sparse Mixture of Convolution Experts (Qiao et al., 2024), DNACHUN-KER learns dynamic segmentation that allocates finer granularity to functional regions (Kim et al., 2026), and other approaches leverage token merging or motif-aware vocabularies (Li et al., 2025; Zhou et al., 2025).

**Architecture and Training.** While BERT-based encoders provide bidirectional context through self-attention (Devlin et al., 2019), they scale quadratically with sequence length. To enable longer contexts, HyenaDNA introduces implicit convolutions for sub-quadratic scaling up to 1 million base pairs (Nguyen et al., 2023), and Caduceus leverages Mamba blocks for efficient bidirectional processing with reverse-complement equivariance (Schiff et al., 2024). JanusDNA combines Mamba, Attention, and Mixture of Experts for million-base-pair contexts (Duan et al., 2025). Training objectives vary: encoder-based GLMs typically use masked language modeling (MLM), while decoder-based models like EVO use autoregressive prediction for sequence generation (Nguyen et al., 2024).

### 2.2. Genomic Annotations as Prior Knowledge

Over the decades, there has been enormous effort in generating functional annotations for the human genome by large-scale research consortia. GENCODE provides comprehensive structural annotations including genes, transcripts, and detailed exon, CDS, and UTR with detailed boundaries, covering approximately 67% of the genome (Mudge et al., 2025). ENCODE catalogs regulatory elements through systematic biochemical assays, identifying candidate Cis-Regulatory Elements(cCREs) including promoter-like sequences (PLS), proximal/distal enhancer-like sequences (pELS, dELS), and CTCF-binding sites (Moore et al., 2020). Regulatory annotations cover approximately 8% of the genome but are critical for interpreting non-coding variants, which comprise the majority of disease-associated genetic variation.

Despite the availability of these comprehensive annotations, few GLMs incorporate functional information during training. While language models can theoretically learn context-dependent representations, explicitly providing functional annotations may accelerate learning and improve representation quality. In this direction, concurrent work has explored encoding annotations into tokenized representations (Medvedev et al., 2025); our approach instead uses cross-modal knowledge distillation to transfer annotation-aware representations to a sequence-only model.

### 2.3. Knowledge Distillation

Knowledge distillation transfers learned representations from a large teacher model to a smaller student model, enabling model compression without substantial performance degradation (Hinton et al., 2015; Gou et al., 2021). The original approach uses soft labels (teacher's output probabilities) to guide student training (Hinton et al., 2015), while subsequent methods extend distillation to hidden states and attention distributions (Sanh et al., 2019; Jiao et al., 2020). These approaches have proven effective for compressing large language models while preserving task performance.

In the genomic domain, recent work has applied distillation to compress DNA language models (Yang et al., 2025). However, existing approaches focus on distillation between models with identical input representations—both teacher and student receive sequence-only input. Our work differs by introducing cross-modal distillation: an annotation-aware teacher transfers its representations to a sequence-only student via hidden state matching. This enables the student to benefit from functional context learned during teacher training while requiring only nucleotide sequences at inference time.

## 3. The annDNA Framework

### 3.1. Overview

The annDNA framework consists of two stages (Figure 1). In Stage 1, we train annotation-aware models using tokens that combine nucleotides with functional annotations. In Stage 2, we transfer the teacher's representations to a sequence-only student via hidden state matching. The resulting model, annDNA-distilled, requires only nucleotide sequences at inference time.

### 3.2. Stage 1: Annotation-Aware Pre-training

**Annotation-Aware Tokenization.** We propose annotation-aware tokenization that combines each nucleotide with its corresponding genomic annotations to form discrete input tokens. We use the GRCh38 reference genome as the base sequence (Schneider et al., 2017). Structural annotations are extracted from GENCODE v49 (Mudge et al., 2025), including gene, transcript, exon, CDS, UTR, start codon, and stop codon. Regulatory annotations are obtained from ENCODE cCREs (Moore et al., 2020), comprising PLS, pELS, dELS, CTCF-binding sites, and H3K4me3 regions.

For each genomic position, we construct a token by concatenating the nucleotide with its overlapping annotations. This yields three tokenization schemes with increasing annotation complexity:

- **annDNA-seq** (Sequence-only): 10 tokens representing nucleotides (A, T, G, C, N) and special tokens.

- **annDNA-struct** (Sequence + Structural): 59 tokens combining nucleotides with structural annotations (e.g., `A_Exon_CDS`).

- **annDNA-full** (Sequence + Structural + Regulatory): 272 tokens incorporating both structural and regulatory annotations (e.g., `G_Intergenic_dELS`).

When multiple annotations overlap at a single position, all relevant annotations are concatenated. Positions without annotations are represented by the nucleotide alone (Figure 1a). We segment the genome into 1000 bp non-overlapping windows for training.

**Model Architecture.** All three models share the same BERT encoder architecture to isolate the effect of tokenization from architectural differences (Devlin et al., 2019). The encoder consists of 12 Transformer layers with 768 hidden dimensions, 12 attention heads, and a feed-forward dimension of 3072, totaling approximately 86 million parameters. The input representation combines learned token embeddings with positional embeddings. We include GROVER (Sanabria et al., 2024) as an external comparison, as it uses a similar model size ($\sim$86M parameters) and was trained on the same reference genome (GRCh38).

**Pre-training Objective.** We train all models using MLM. For each input sequence, we randomly mask 15% of tokens, replacing 80% with a `[MASK]` token, 10% with a random token, and 10% unchanged. The model predicts the original token at masked positions using a linear classification head. Training uses chromosomes 1–21 and X, with chromosome 22 held out for validation. We train for 10 epochs using AdamW optimizer with learning rate $5 \times 10^{-5}$, batch size 64, and linear warmup followed by cosine decay.

### 3.3. Stage 2: Cross-Modal Knowledge Distillation

**Motivation.** Annotation-aware models require genomic annotations as input, limiting applicability when annotations are unavailable or incomplete. To address this, we employ knowledge distillation via hidden state matching to transfer functional representations from annDNA-full to a sequence-only student model (Figure 1b).

**Student Architecture.** The student receives sequence-only input identical to annDNA-seq. To enable direct hidden state comparison, the student retains the same hidden dimension ($d = 768$) as the teacher. We reduce the number of layers from 12 to 4 and attention heads from 12 to 4, resulting in approximately 28 million parameters—one-third of the teacher.

**Distillation Objective.** We train the student using a combined loss:

$$\mathcal{L} = \alpha \cdot \mathcal{L}_{\text{CE}} + (1 - \alpha) \cdot \mathcal{L}_{\text{MSE}} \qquad (1)$$

where $\mathcal{L}_{\text{CE}}$ is the cross-entropy loss for masked token prediction and $\mathcal{L}_{\text{MSE}}$ is the mean squared error between the final-layer hidden states of teacher and student, computed over non-padding positions. We set $\alpha = 0.5$ to weight both objectives equally, following established practice in language model distillation (Sanh et al., 2019). The teacher's annotation-aware tokens are converted to sequence-only tokens by extracting the nucleotide component (e.g., `A_Exon_CDS_PLS` $\rightarrow$ `A`). Training uses AdamW optimizer with learning rate $5 \times 10^{-5}$, batch size 128, and linear warmup over 1000 steps followed by cosine decay.

We quantify distillation effectiveness using accuracy gap recovery:

$$\text{Recovery} = \frac{\text{Acc}_{\text{distilled}} - \text{Acc}_{\text{seq}}}{\text{Acc}_{\text{full}} - \text{Acc}_{\text{seq}}} \qquad (2)$$

### 3.4. Evaluation Protocol

We evaluate model performance through three complementary analyses.

**Genomic Region Embedding.** To assess whether models learn biologically meaningful representations, we extract embeddings from genomic regions with known functional annotations and measure their separability. We sample sequences from the GRCh38 reference genome based on annDNA-full's tokenization: regions are identified by scanning chromosome tokens for contiguous stretches where $\geq$50% of positions match the target category, with a minimum 10 kb gap between samples to avoid overlap. We sample up to 500 bp sequences from four structural categories (CDS, UTR, intron, intergenic) and five regulatory categories (promoter, pELS, dELS, CTCF-binding site, H3K4me3), with up to 1,000 samples per category balanced across training chromosomes (chr1–21, chrX) and validation chromosome (chr22; Appendix Table 6). For each sequence, we obtain embeddings by mean pooling the last hidden states across non-padding tokens. We quantify embedding quality using linear probing—training a logistic regression classifier on the embeddings with 5-fold cross-validation across 3 random seeds, reporting mean AUROC.

**Variant Effect Prediction.** We evaluate variant representations using four benchmark datasets (Appendix Table 7). ClinVar pathogenic variants use pathogenic-labeled variants as positives and common variants (MAF > 5% in gnomAD) as negatives, preprocessed by GPN-MSA (Benegas et al., 2025a). GTEx eQTL variants use statistically fine-mapped causal variants (posterior inclusion probability >

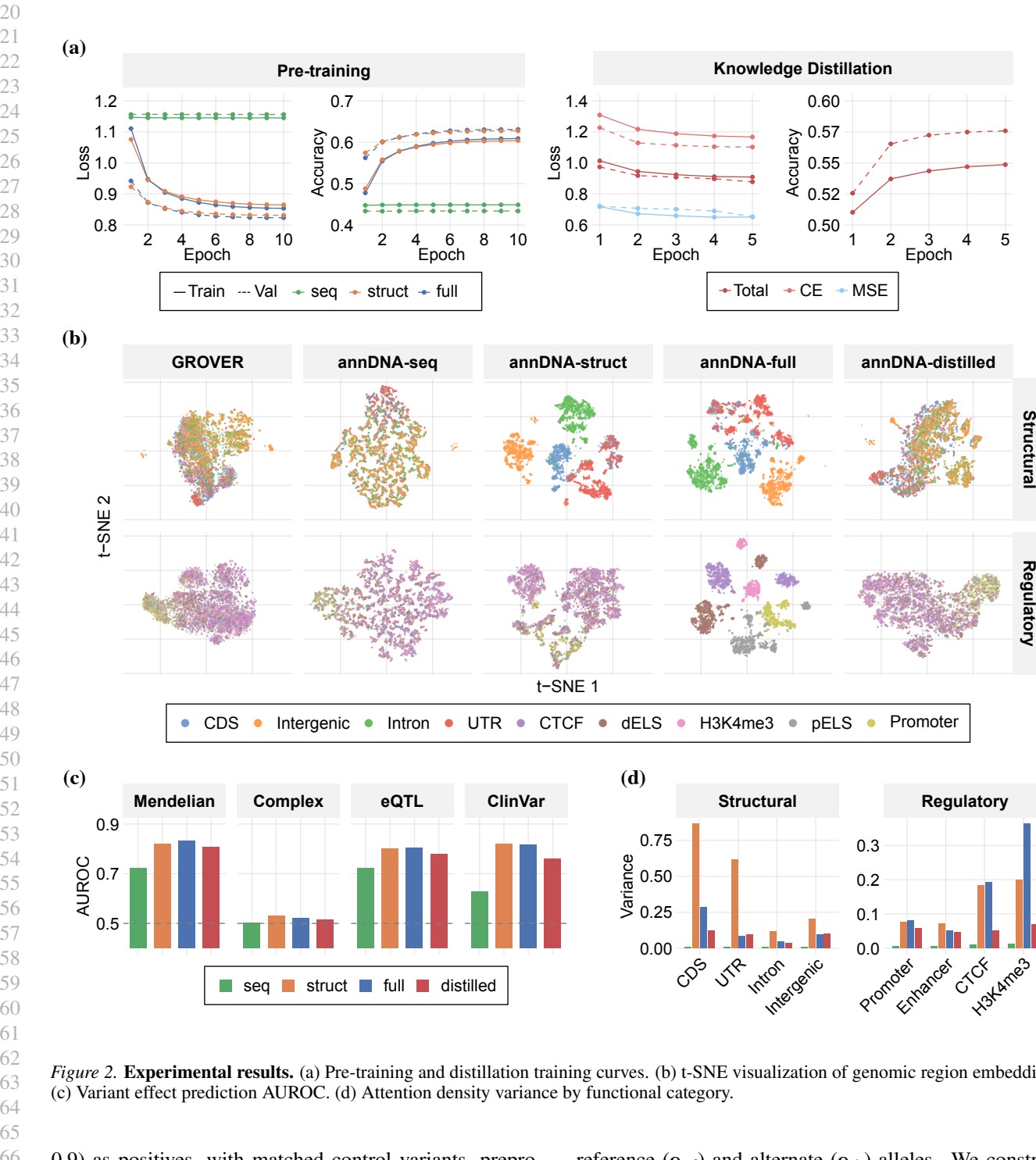

*Figure 2.* **Experimental results.** (a) Pre-training and distillation training curves. (b) t-SNE visualization of genomic region embeddings. (c) Variant effect prediction AUROC. (d) Attention density variance by functional category.

0.9) as positives, with matched control variants, preprocessed by Enformer (Avsec et al., 2021). TraitGym provides Mendelian variants from OMIM and Complex trait variants from GWAS fine-mapping (Benegas et al., 2025b). We test multiple window sizes (250, 500, 750 bp) and variant positions (0.25, 0.5, 0.75), with main results reported at 500 bp window and position 0.5 (full sensitivity analysis in Appendix). For each variant, we extract embeddings for both

reference ($\mathbf{e}_{\text{ref}}$) and alternate ($\mathbf{e}_{\text{alt}}$) alleles. We construct features by concatenating:

$$\mathbf{f} = [\mathbf{e}_{\text{ref}}; \mathbf{e}_{\text{alt}}; \mathbf{e}_{\text{alt}} - \mathbf{e}_{\text{ref}}; \mathbf{e}_{\text{ref}} \odot \mathbf{e}_{\text{alt}}; |\mathbf{e}_{\text{alt}} - \mathbf{e}_{\text{ref}}|] \quad (3)$$

where $\odot$ denotes element-wise product. We apply the same linear probing procedure to distinguish causal from control variants, reporting AUROC.

**Attention Analysis.** To understand how models allocate attention across functional elements, we analyze attention patterns using annDNA-full's annotation scheme as reference labels for all models. We select 300 genomic windows containing multiple distinct functional categories (Appendix Table 8). For each window, we extract attention weights from all layers and heads and compute attention density for each functional category $c$:

$$D_c^{(l,h)} = \frac{1}{|P_c|} \sum_{j \in P_c} \sum_{i=1}^{n} A_{ij}^{(l,h)} \tag{4}$$

where $P_c$ is the set of positions belonging to category $c$, $A^{(l,h)}$ is the attention matrix at layer $l$ and head $h$, and $n$ is the sequence length. We report variance in attention density across samples—higher variance indicates differential attention to functional elements, while variance near zero indicates uniform attention (see Appendix Figure 3 for heatmap visualization).

# 4. Experiments

## 4.1. Model Training

Pre-training performance varied depending on input complexity (Figure 2a, left). annDNA-seq showed limited learning capacity with a validation accuracy of 0.434, suggesting that predicting masked tokens from a simple nucleotide vocabulary provides a weak supervision signal. In contrast, annDNA-struct increased validation accuracy to 0.627, and annDNA-full further improved it to 0.631.

In the knowledge distillation phase, annDNA-distilled demonstrated efficient convergence (Figure 2a, right). Despite receiving the same sequence-only input as annDNA-seq, annDNA-distilled achieved a final validation accuracy of 0.576 by minimizing both cross-entropy and feature-matching MSE losses. These results indicate that annDNA-distilled effectively captures the teacher's representations, recovering 72% of the performance gap between annDNA-seq and annDNA-full. Figure 1c summarizes these results, showing that annotation-aware models outperform the sequence-only baseline while annDNA-distilled achieves competitive performance with only one-third of the parameters.

## 4.2. Downstream Evaluation

**Genomic Region Embedding.** We evaluated the quality of learned representations by quantifying class separability in the embedding space (Table 1).

These embeddings showed increasing separation of genomic regions from annDNA-seq to annDNA-full (Figure 2b). annDNA-seq embeddings exhibit substantial overlap across functional categories, reflecting a lack of distinct biological context. Conversely, annDNA-full displays clearer separa-

*Table 1.* Linear probing AUROC for genomic region classification. **Bold**: best, underline: second best. Gain: annDNA-distilled − annDNA-seq.

| Model | Structural | Regulatory |
|-------|-----------|-----------|
| GROVER | .7824 ±.0030 | .5843 ±.0043 |
| annDNA-seq | .7643 ±.0009 | .6071 ±.0020 |
| annDNA-struct | **.9912** ±.0005 | .6859 ±.0023 |
| annDNA-full | .9889 ±.0004 | **.9981** ±.0001 |
| annDNA-distilled | .8363 ±.0004 | .6314 ±.0043 |
| **Gain** | **+.0720** | **+.0243** |

tions within structural and regulatory categories. Notably, annDNA-struct—which has no regulatory annotations—still achieves better regulatory classification (AUROC 0.686) than annDNA-seq (0.607), suggesting that structural context provides indirect signals about regulatory functions. annDNA-distilled also exhibits noticeably clearer boundaries than annDNA-seq, particularly for structural regions. annDNA-distilled surpasses not only annDNA-seq (+0.072 structural, +0.024 regulatory) but also GROVER (+0.054 structural, +0.047 regulatory), demonstrating that distillation successfully transfers the teacher's annotation-aware representations.

*Table 2.* Variant effect prediction AUROC. **Bold**: best, underline: second best. Gain: -distilled − -seq. annDNA model names are abbreviated (e.g., -seq for annDNA-seq).

| Model | ClinVar | eQTL | Mendel. | Complex |
|-------|---------|------|---------|---------|
| GROVER | .7282±.0009 | .7771±.0005 | **.8689**±.0035 | .5250±.0013 |
| -seq | .6276±.0004 | .7236±.0006 | .7224±.0085 | .5021±.0080 |
| -struct | **.8183**±.0001 | .8002±.0008 | .8187±.0009 | **.5306**±.0027 |
| -full | .8151±.0001 | **.8043**±.0008 | .8337±.0084 | .5220±.0028 |
| -distilled | .7608±.0001 | .7794±.0005 | .8080±.0054 | .5149±.0056 |
| **Gain** | **+.1332** | **+.0558** | **+.0856** | **+.0128** |

**Variant Effect Prediction.** We further assessed the practical utility of these representations using variant effect prediction benchmarks (Table 2). Using the default configuration (500 bp window, variant at position 0.5), annotation-aware models achieved higher performance than the sequence-only model(Figure 2c). annDNA-struct and annDNA-full consistently outperformed annDNA-seq across all tasks; specifically, for ClinVar pathogenic variants, annDNA-struct achieved the highest AUROC of 0.8183 compared to annDNA-seq's 0.6276. For regulatory variants (eQTL), annDNA-full achieved the highest performance (0.8043), demonstrating that regulatory annotations are informative for expression-related variants.

Notably, annDNA-distilled outperformed annDNA-seq, achieving gains of +0.1332 on ClinVar and +0.0856 on Mendelian benchmarks. This confirms that the distillation

process successfully injected functional priors that the student could not learn from the sequence modeling objective alone. Moreover, annDNA-distilled surpassed the external baseline GROVER on ClinVar (+0.033) and eQTL (+0.002) tasks. This indicates that distillation from annotation-aware teachers yields stronger variant representations than conventional sequence-only pre-training, providing a highly effective strategy for model development.

*Table 3.* Attention density variance (upper: structural, lower: regulatory). **Bold**: best, underline: second best. Gain: -distilled − -seq. annDNA model names are abbreviated.

| Region | -seq | -struct | -full | -distilled | **Gain** |
|---|---|---|---|---|---|
| CDS | 0.0107 | **0.8633** | 0.2876 | 0.1224 | **+0.1117** |
| UTR | 0.0073 | **0.6146** | 0.0848 | 0.0957 | **+0.0884** |
| Intron | 0.0059 | **0.1169** | 0.0445 | 0.0356 | **+0.0297** |
| Intergenic | 0.0100 | **0.2027** | 0.0961 | 0.0980 | **+0.0880** |
| Promoter | 0.0053 | 0.0772 | **0.0806** | 0.0582 | **+0.0529** |
| Enhancer | 0.0053 | **0.0710** | 0.0501 | 0.0468 | **+0.0415** |
| CTCF | 0.0102 | 0.1832 | **0.1923** | 0.0515 | **+0.0413** |
| H3K4me3 | 0.0116 | 0.1984 | **0.3626** | 0.0702 | **+0.0586** |

**Attention Analysis.** Finally, attention density variance analysis provides insight into the biological features prioritized by each model (Figure 2d, Table 3). High variance indicates selective attention that differentiates positions within that category, while variance near zero indicates uniform attention across all positions.

annDNA-seq exhibits a flat distribution with near-zero variance across all categories. In contrast, annDNA-struct shows strong attention peaks specifically at structural elements (CDS, UTR), while annDNA-full distributes attention to both structural and regulatory regions (e.g., H3K4me3, CTCF; Figure 3). annDNA-distilled mimics this behavior, showing elevated variance across all functional categories compared to annDNA-seq—both structural (CDS, UTR, intron, intergenic) and regulatory (promoter, enhancer, CTCF, H3K4me3). As shown in Table 3, annDNA-distilled achieves consistent gains over annDNA-seq across all regions, confirming that it has learned to attend to biologically relevant positions even without explicit input annotations.

## 5. Conclusion

We demonstrated that incorporating genomic annotations into input representations improves the representation power of GLMs. Our experiments showed consistent performance improvements from annDNA-seq to annDNA-full across multiple evaluation tasks. In genomic region embedding analysis, annDNA-full embeddings exhibit increased separability for regulatory elements. For variant effect prediction, annotation-aware models outperformed sequence-only baselines across all benchmarks. Attention analysis revealed that

annotation-aware models develop focused attention patterns on functional elements, while annDNA-seq shows uniform attention across all positions.

We further showed that knowledge distillation enables the transfer of annotation-aware representations to a sequence-only student model. annDNA-distilled outperformed annDNA-seq across all benchmarks despite using identical input, achieving 72% of the accuracy gap recovery in pre-training. With only one-third of the teacher's parameters (28M vs. 86M), annDNA-distilled achieved intermediate performance between annDNA-seq and annDNA-struct/annDNA-full, demonstrating that functional representations can be learned without explicit annotations. Notably, annDNA-distilled also exhibited elevated attention variance on structural and regulatory elements compared to annDNA-seq, confirming successful knowledge transfer even in attention patterns. This enables practical deployment in settings where genomic annotations are unavailable or computationally expensive to generate.

Our results suggest that the path forward for GLMs lies not only in scaling architectures, but in rethinking how biological sequences are represented. By encoding functional context at the input level, models can learn genomic organization more effectively. Moreover, knowledge distillation provides a practical mechanism for deploying such models in settings where annotations are unavailable or costly to obtain.

## 6. Limitations and Future Work

Our work has several limitations that suggest directions for future research. First, we held out chromosome 22 for validation; incorporating it and additional chromosomes for training may improve generalization. Second, extending to multi-species genomes could leverage evolutionary conservation for improved representations. Third, our current annotation scheme could be enriched with isoform-level information and cell-type-specific regulatory annotations from ENCODE, which may further improve tissue-specific variant effect prediction. Fourth, our models use 1000 bp windows with standard BERT architecture. Scaling to longer contexts using efficient architectures such as attention U-Net in NTv3 (Boshar et al., 2025) or StripedHyena in Evo (Nguyen et al., 2024) could model distal regulatory interactions. Since our annotation-aware tokenization is model-agnostic, combining it with these architectural innovations for long-range modeling could yield further improvements.

## Impact Statement

This work enhances the interpretability of non-coding variants by integrating functional annotations, which can con-

tribute to genetic disease diagnosis and drug target discovery. Our knowledge distillation approach produces a lightweight model, lowering computational requirements and supporting more resource-efficient deployment. However, since our models rely on the GRCh38 reference genome and existing annotation sets, they may carry inherent biases. Users should ensure validation across diverse populations for clinical applications.

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

## A. Data Statistics

We utilized the GRCh38 reference genome to construct our annotation-aware tokens. Structural annotations were obtained from GENCODE v49 (`https://www.gencodegenes.org/human/`), which provides comprehensive gene feature definitions including exons and CDS. Regulatory elements were sourced from the ENCODE SCREEN database (`https://screen.encodeproject.org/`), covering cCREs. Table 4 summarizes genomic annotation coverage, and Table 5 shows the resulting vocabulary for each model.

*Table 4.* Genomic annotation statistics from GRCh38, GENCODE v49, and ENCODE cCREs.

| Type | Feature | Total bp | Coverage |
|------|---------|----------|----------|
| Reference | GRCh38 | 3,088.3M | 100% |
| Structural | Gene | 2,064.2M | 66.84% |
| | Transcript | 2,064.2M | 66.84% |
| | Exon | 191.8M | 6.21% |
| | CDS | 36.5M | 1.18% |
| | UTR | 68.6M | 2.22% |
| | Start codon | 0.09M | 0.00% |
| | Stop codon | 0.14M | 0.00% |
| Regulatory | PLS | 10.0M | 0.32% |
| | pELS | 36.6M | 1.19% |
| | dELS | 185.7M | 6.01% |
| | CTCF | 14.3M | 0.46% |
| | H3K4me3 | 6.8M | 0.22% |

*Table 5.* Token vocabulary for each model.

| Model | Vocab. Size | Example Tokens |
|-------|-------------|----------------|
| annDNA-seq | 10 | A, T, G, C, N |
| annDNA-struct | 59 | A_Exon_CDS, T_Intron, G_UTR |
| annDNA-full | 272 | A_Exon_CDS_PLS, G_dELS |

## B. Evaluation Dataset Statistics

Table 6 summarizes the genomic region samples for embedding analysis. Regions were identified by scanning chromosome tokens for contiguous stretches where ≥50% of positions matched the target category. Non-overlapping samples were selected with a minimum 10 kb gap, balanced across categories up to 1,000 per category. Training samples were drawn from chr1–21 and chrX, validation from chr22. Table 7 provides sample counts for each variant effect prediction benchmark. The benchmark datasets were retrieved from public repositories to ensure reproducibility: the fine-mapped GTEx eQTL data is available via the Enformer Google Cloud storage (`https://console.cloud.google.com/storage/browser/dm-enformer/data/gtex_fine`), and the ClinVar dataset preprocessed for GPN-MSA is hosted on Hugging Face (`https://huggingface.co/datasets/songlab/clinvar`). Table 8 describes the attention analysis samples. Samples were selected from training chromosomes (chr1–21, chrX) by scanning tokenized sequences using 1,000 bp windows with 10,000 bp stride. Each window was required to contain at least 4 distinct functional categories, with windows containing >10% ambiguous bases excluded. Category presence was determined using coverage thresholds: structural categories required ≥50 positions, regulatory categories (promoter, enhancer) required ≥10 positions, and rare elements (CTCF, H3K4me3) required ≥1 position. Windows were ranked by diversity score with priority for rare regulatory elements, and the top 300 windows were selected.

*Table 6.* Genomic region embedding dataset.

| Type | Category | Train | Val |
|------|----------|-------|-----|
| Structural | CDS | 1,000 | 43 |
| | UTR | 1,000 | 52 |
| | Intron | 1,000 | 43 |
| | Intergenic | 1,000 | 85 |
| Regulatory | Promoter | 1,000 | 32 |
| | pELS | 1,000 | 43 |
| | dELS | 1,000 | 51 |
| | CTCF | 1,000 | 34 |
| | H3K4me3 | 1,000 | 17 |

*Table 7.* Variant effect prediction benchmarks.

| Dataset | Positive | Negative | Total |
|---------|----------|----------|-------|
| ClinVar | 21,942 | 18,260 | 40,202 |
| eQTL | 19,779 | 20,047 | 39,826 |
| Mendelian | 336 | 3,024 | 3,360 |
| Complex | 1,112 | 10,008 | 11,120 |

*Table 8.* Attention analysis samples: 300 windows of 1,000 bp containing diverse functional categories.

| Type | Category | Count |
|------|----------|-------|
| Structural | CDS | 255 |
| | UTR | 291 |
| | Intron | 291 |
| | Intergenic | 200 |
| Regulatory | Promoter | 277 |
| | Enhancer | 290 |
| | CTCF | 5 |
| | H3K4me3 | 20 |

## C. Full Embedding Results

Table 9 shows per-category linear probing AUROC for train and validation splits.

*Table 9.* Per-category linear probing AUROC. **Bold**: best, underline: second best. Gain: -distilled − -seq. annDNA model names are abbreviated.

| Category | Split | GROVER | -seq | -struct | -full | -distilled | Gain |
|---|---|---|---|---|---|---|---|
| CDS | Train | $.8277_{\pm.0014}$ | $.8014_{\pm.0026}$ | $\mathbf{.9977}_{\pm.0003}$ | $\underline{.9968}_{\pm.0003}$ | $.8746_{\pm.0011}$ | +.0732 |
|  | Val | $.7958_{\pm.0312}$ | $.7678_{\pm.0230}$ | $\underline{.9740}_{\pm.0132}$ | $\mathbf{.9762}_{\pm.0098}$ | $.8449_{\pm.0313}$ | +.0771 |
| UTR | Train | $.7685_{\pm.0013}$ | $.7414_{\pm.0021}$ | $\mathbf{.9867}_{\pm.0003}$ | $\underline{.9838}_{\pm.0004}$ | $.8207_{\pm.0014}$ | +.0793 |
|  | Val | $.7296_{\pm.0173}$ | $.7119_{\pm.0311}$ | $\mathbf{.9712}_{\pm.0103}$ | $\underline{.9599}_{\pm.0107}$ | $.7913_{\pm.0277}$ | +.0794 |
| Intron | Train | $.7746_{\pm.0012}$ | $.7577_{\pm.0012}$ | $\mathbf{.9923}_{\pm.0002}$ | $\underline{.9908}_{\pm.0003}$ | $.8324_{\pm.0010}$ | +.0747 |
|  | Val | $.7571_{\pm.0257}$ | $.7260_{\pm.0298}$ | $\mathbf{.9884}_{\pm.0088}$ | $\underline{.9862}_{\pm.0071}$ | $.8102_{\pm.0318}$ | +.0842 |
| Intergenic | Train | $.7606_{\pm.0011}$ | $.7539_{\pm.0019}$ | $\underline{.9880}_{\pm.0002}$ | $\mathbf{.9885}_{\pm.0002}$ | $.8275_{\pm.0009}$ | +.0736 |
|  | Val | $.8484_{\pm.0160}$ | $.8564_{\pm.0174}$ | $\mathbf{.9993}_{\pm.0011}$ | $\underline{.9982}_{\pm.0012}$ | $.8924_{\pm.0151}$ | +.0360 |
| Promoter | Train | $.5593_{\pm.0078}$ | $.5853_{\pm.0064}$ | $\underline{.6574}_{\pm.0051}$ | $\mathbf{.9997}_{\pm.0002}$ | $.6223_{\pm.0046}$ | +.0370 |
|  | Val | $.5340_{\pm.0240}$ | $.5595_{\pm.0326}$ | $\underline{.6420}_{\pm.0319}$ | $\mathbf{1.000}_{\pm.0000}$ | $.5847_{\pm.0247}$ | +.0252 |
| dELS | Train | $.5970_{\pm.0045}$ | $.6262_{\pm.0038}$ | $\underline{.7017}_{\pm.0045}$ | $\mathbf{.9990}_{\pm.0004}$ | $.6542_{\pm.0044}$ | +.0280 |
|  | Val | $.5682_{\pm.0078}$ | $.6163_{\pm.0144}$ | $\underline{.6747}_{\pm.0171}$ | $\mathbf{.9998}_{\pm.0003}$ | $.6498_{\pm.0227}$ | +.0335 |
| pELS | Train | $.5707_{\pm.0111}$ | $.5721_{\pm.0253}$ | $\underline{.6557}_{\pm.0163}$ | $\mathbf{.9995}_{\pm.0007}$ | $.6127_{\pm.0109}$ | +.0406 |
|  | Val | $.5092_{\pm.0200}$ | $.5228_{\pm.0142}$ | $\underline{.5952}_{\pm.0214}$ | $\mathbf{.9990}_{\pm.0011}$ | $.5146_{\pm.0234}$ | −.0082 |
| CTCF | Train | $.6405_{\pm.0179}$ | $.6649_{\pm.0152}$ | $\underline{.7136}_{\pm.0082}$ | $\mathbf{.9989}_{\pm.0018}$ | $.6977_{\pm.0180}$ | +.0328 |
|  | Val | $.5639_{\pm.0256}$ | $.5798_{\pm.0244}$ | $\underline{.6826}_{\pm.0308}$ | $\mathbf{1.000}_{\pm.0000}$ | $.6180_{\pm.0241}$ | +.0382 |
| H3K4me3 | Train | $.5159_{\pm.0370}$ | $\underline{.5784}_{\pm.0100}$ | $.5981_{\pm.0126}$ | $\mathbf{.9978}_{\pm.0038}$ | $.5729_{\pm.0170}$ | −.0055 |
|  | Val | $.5233_{\pm.0481}$ | $.4942_{\pm.0607}$ | $\underline{.6011}_{\pm.0613}$ | $\mathbf{.9929}_{\pm.0143}$ | $.5352_{\pm.0324}$ | +.0410 |

# D. Benchmark Sensitivity Analysis

Table 10 shows variant effect prediction AUROC across window sizes (250, 500, 750 bp) and variant positions within the window (0.25, 0.5, 0.75).

*Table 10.* Variant effect prediction AUROC across configurations. **Bold**: best, underline: second best. Gain: annDNA-distilled − annDNA-seq.

| Dataset | Win. | Pos. | GROVER | -seq | -struct | -full | -distilled | Gain |
|---|---|---|---|---|---|---|---|---|
| ClinVar | 250 | 0.25 | $.7021_{\pm.0003}$ | $.6181_{\pm.0003}$ | $\underline{.7966}_{\pm.0002}$ | $\mathbf{.8033}_{\pm.0003}$ | $.7388_{\pm.0003}$ | +.1207 |
|  | 250 | 0.50 | $.7096_{\pm.0008}$ | $.6201_{\pm.0009}$ | $\underline{.7979}_{\pm.0001}$ | $\mathbf{.8063}_{\pm.0013}$ | $.7410_{\pm.0006}$ | +.1209 |
|  | 250 | 0.75 | $.7006_{\pm.0006}$ | $.6119_{\pm.0012}$ | $\underline{.7953}_{\pm.0004}$ | $\mathbf{.7963}_{\pm.0003}$ | $.7321_{\pm.0002}$ | +.1202 |
|  | 500 | 0.25 | $.7235_{\pm.0002}$ | $.6240_{\pm.0013}$ | $\mathbf{.8201}_{\pm.0004}$ | $\underline{.8120}_{\pm.0002}$ | $.7517_{\pm.0005}$ | +.1277 |
|  | 500 | 0.50 | $.7282_{\pm.0009}$ | $.6276_{\pm.0004}$ | $\mathbf{.8183}_{\pm.0001}$ | $\underline{.8151}_{\pm.0001}$ | $.7608_{\pm.0001}$ | +.1332 |
|  | 500 | 0.75 | $.7221_{\pm.0002}$ | $.6265_{\pm.0007}$ | $\mathbf{.8170}_{\pm.0006}$ | $\underline{.8119}_{\pm.0007}$ | $.7538_{\pm.0004}$ | +.1273 |
|  | 750 | 0.25 | $.7316_{\pm.0008}$ | $.6272_{\pm.0016}$ | $\mathbf{.8212}_{\pm.0003}$ | $\underline{.8149}_{\pm.0003}$ | $.7596_{\pm.0009}$ | +.1324 |
|  | 750 | 0.50 | $.7354_{\pm.0005}$ | $.6353_{\pm.0005}$ | $\mathbf{.8255}_{\pm.0005}$ | $\underline{.8184}_{\pm.0007}$ | $.7661_{\pm.0003}$ | +.1308 |
|  | 750 | 0.75 | $.7301_{\pm.0007}$ | $.6297_{\pm.0007}$ | $\mathbf{.8248}_{\pm.0005}$ | $\underline{.8166}_{\pm.0005}$ | $.7620_{\pm.0003}$ | +.1323 |
| Complex | 250 | 0.25 | $.5258_{\pm.0027}$ | $.5125_{\pm.0059}$ | $\mathbf{.5310}_{\pm.0056}$ | $.5158_{\pm.0038}$ | $\underline{.5234}_{\pm.0053}$ | +.0109 |
|  | 250 | 0.50 | $\mathbf{.5302}_{\pm.0042}$ | $.5061_{\pm.0010}$ | $\underline{.5235}_{\pm.0073}$ | $.5219_{\pm.0042}$ | $.5112_{\pm.0042}$ | +.0051 |
|  | 250 | 0.75 | $\underline{.5202}_{\pm.0030}$ | $.5017_{\pm.0059}$ | $.5143_{\pm.0020}$ | $\mathbf{.5159}_{\pm.0054}$ | $.5074_{\pm.0007}$ | +.0057 |
|  | 500 | 0.25 | $.5235_{\pm.0058}$ | $\underline{.5178}_{\pm.0011}$ | $\mathbf{.5262}_{\pm.0024}$ | $.5158_{\pm.0075}$ | $.5089_{\pm.0057}$ | −.0089 |
|  | 500 | 0.50 | $.5250_{\pm.0013}$ | $.5021_{\pm.0080}$ | $\mathbf{.5306}_{\pm.0027}$ | $\underline{.5220}_{\pm.0028}$ | $.5149_{\pm.0056}$ | +.0128 |
|  | 500 | 0.75 | $.5240_{\pm.0059}$ | $.5092_{\pm.0038}$ | $\mathbf{.5313}_{\pm.0020}$ | $\underline{.5185}_{\pm.0037}$ | $.5163_{\pm.0057}$ | +.0071 |
|  | 750 | 0.25 | $.5211_{\pm.0052}$ | $.5212_{\pm.0091}$ | $\underline{.5328}_{\pm.0034}$ | $\mathbf{.5330}_{\pm.0076}$ | $.5274_{\pm.0053}$ | +.0062 |
|  | 750 | 0.50 | $.5244_{\pm.0038}$ | $.5134_{\pm.0059}$ | $\mathbf{.5383}_{\pm.0048}$ | $\underline{.5244}_{\pm.0043}$ | $.5107_{\pm.0061}$ | −.0027 |
|  | 750 | 0.75 | $.5088_{\pm.0066}$ | $.5024_{\pm.0017}$ | $\mathbf{.5291}_{\pm.0013}$ | $.5219_{\pm.0057}$ | $\underline{.5227}_{\pm.0069}$ | +.0203 |
| eQTL | 250 | 0.25 | $.7712_{\pm.0005}$ | $.7230_{\pm.0006}$ | $\underline{.7931}_{\pm.0007}$ | $\mathbf{.7987}_{\pm.0005}$ | $.7729_{\pm.0005}$ | +.0499 |
|  | 250 | 0.50 | $.7766_{\pm.0013}$ | $.7220_{\pm.0008}$ | $\underline{.7949}_{\pm.0007}$ | $\mathbf{.8024}_{\pm.0007}$ | $.7751_{\pm.0007}$ | +.0531 |
|  | 250 | 0.75 | $.7717_{\pm.0008}$ | $.7226_{\pm.0010}$ | $\underline{.7937}_{\pm.0008}$ | $\mathbf{.8002}_{\pm.0002}$ | $.7698_{\pm.0009}$ | +.0472 |
|  | 500 | 0.25 | $.7804_{\pm.0006}$ | $.7206_{\pm.0007}$ | $\underline{.7970}_{\pm.0005}$ | $\mathbf{.8032}_{\pm.0002}$ | $.7803_{\pm.0005}$ | +.0597 |

**Table 10 – continued from previous page**

| Dataset | Win. | Pos. | GROVER | -seq | -struct | -full | -distilled | Gain |
|---|---|---|---|---|---|---|---|---|
| | 500 | 0.50 | $.7771_{\pm.0005}$ | $.7236_{\pm.0006}$ | $\underline{.8002}_{\pm.0008}$ | $\mathbf{.8043}_{\pm.0008}$ | $.7794_{\pm.0005}$ | +.0558 |
| | 500 | 0.75 | $.7765_{\pm.0004}$ | $.7219_{\pm.0004}$ | $\mathbf{.8004}_{\pm.0004}$ | $\underline{.8000}_{\pm.0004}$ | $.7796_{\pm.0004}$ | +.0577 |
| | 750 | 0.25 | $.7809_{\pm.0011}$ | $.7173_{\pm.0005}$ | $\underline{.8009}_{\pm.0002}$ | $\mathbf{.8030}_{\pm.0005}$ | $.7800_{\pm.0001}$ | +.0627 |
| | 750 | 0.50 | $.7800_{\pm.0009}$ | $.7202_{\pm.0001}$ | $\underline{.8004}_{\pm.0001}$ | $\mathbf{.8037}_{\pm.0012}$ | $.7803_{\pm.0003}$ | +.0601 |
| | 750 | 0.75 | $.7769_{\pm.0008}$ | $.7245_{\pm.0002}$ | $\underline{.8009}_{\pm.0006}$ | $\mathbf{.8041}_{\pm.0002}$ | $.7775_{\pm.0004}$ | +.0530 |
| Mendelian | 250 | 0.25 | $\mathbf{.8610}_{\pm.0020}$ | $.6847_{\pm.0066}$ | $\underline{.8100}_{\pm.0047}$ | $.7994_{\pm.0012}$ | $.7913_{\pm.0056}$ | +.1066 |
| | 250 | 0.50 | $\mathbf{.8494}_{\pm.0079}$ | $.6875_{\pm.0049}$ | $.7867_{\pm.0048}$ | $\underline{.8072}_{\pm.0036}$ | $.7980_{\pm.0042}$ | +.1105 |
| | 250 | 0.75 | $\mathbf{.8361}_{\pm.0042}$ | $.6838_{\pm.0046}$ | $\underline{.8008}_{\pm.0071}$ | $.8154_{\pm.0068}$ | $.7975_{\pm.0077}$ | +.1137 |
| | 500 | 0.25 | $\mathbf{.8508}_{\pm.0049}$ | $.7176_{\pm.0017}$ | $.8161_{\pm.0040}$ | $\underline{.8242}_{\pm.0053}$ | $.8153_{\pm.0054}$ | +.0977 |
| | 500 | 0.50 | $\mathbf{.8689}_{\pm.0035}$ | $.7224_{\pm.0085}$ | $.8187_{\pm.0009}$ | $\underline{.8337}_{\pm.0084}$ | $.8080_{\pm.0054}$ | +.0856 |
| | 500 | 0.75 | $\mathbf{.8622}_{\pm.0049}$ | $.7426_{\pm.0065}$ | $.8253_{\pm.0037}$ | $\underline{.8439}_{\pm.0023}$ | $.8158_{\pm.0013}$ | +.0732 |
| | 750 | 0.25 | $\mathbf{.8633}_{\pm.0037}$ | $.7492_{\pm.0046}$ | $.8096_{\pm.0049}$ | $\underline{.8448}_{\pm.0133}$ | $.8396_{\pm.0076}$ | +.0904 |
| | 750 | 0.50 | $\mathbf{.8436}_{\pm.0033}$ | $.7483_{\pm.0018}$ | $\underline{.8307}_{\pm.0016}$ | $.8435_{\pm.0012}$ | $.8186_{\pm.0058}$ | +.0703 |
| | 750 | 0.75 | $\mathbf{.8727}_{\pm.0026}$ | $.7516_{\pm.0029}$ | $.8280_{\pm.0042}$ | $\underline{.8669}_{\pm.0026}$ | $.8191_{\pm.0073}$ | +.0675 |

# E. Attention Visualization

Figure 3 visualizes attention patterns for annDNA-seq, annDNA-full, and annDNA-distilled. annDNA-seq shows uniform attention across all positions, while annDNA-full focuses sharply on functional elements. annDNA-distilled successfully replicates this focused pattern despite lacking annotation input.

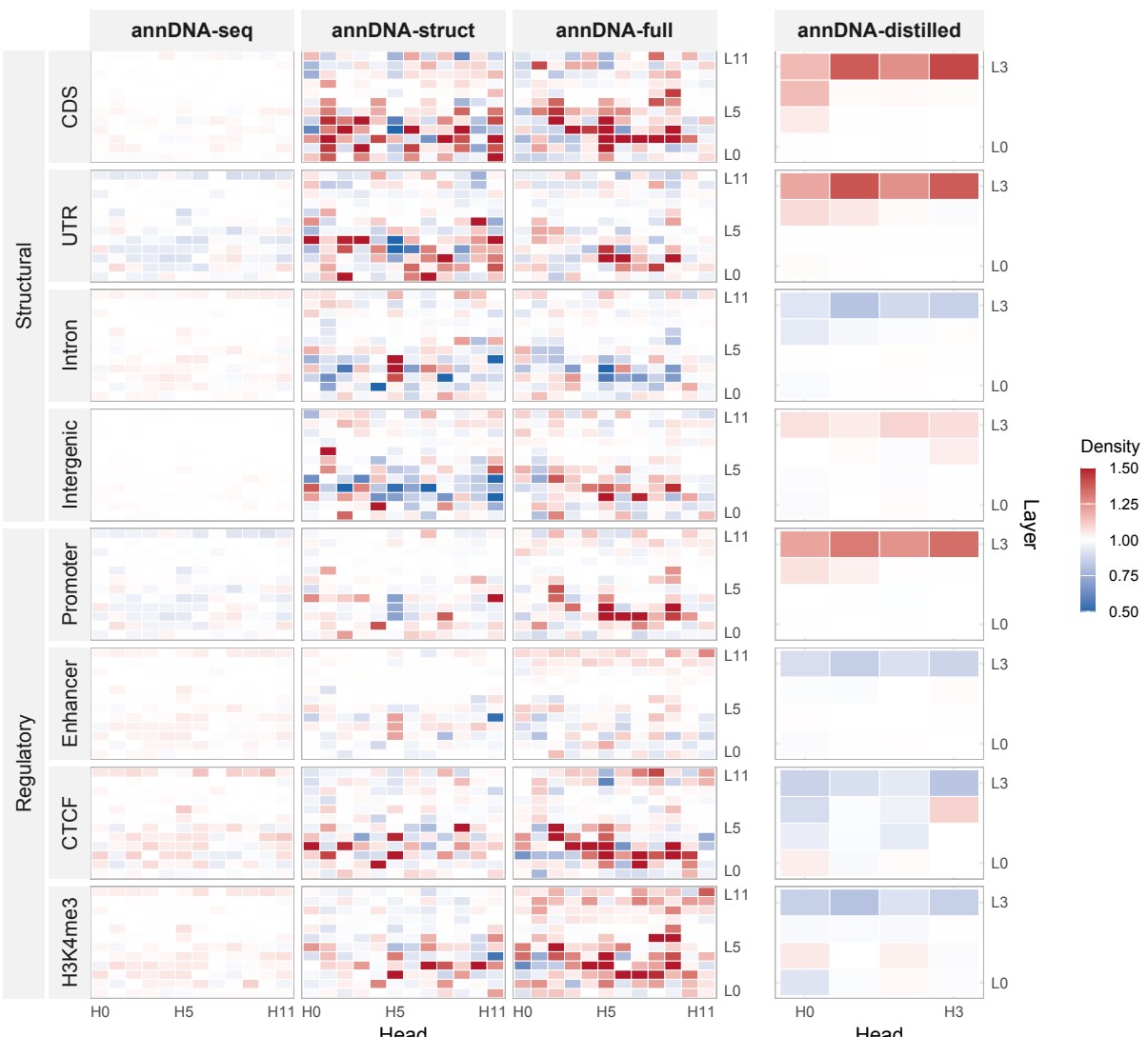

*Figure 3.* Attention density heatmap.

