# OpenReview forum: "annDNA: Learning Annotation-Aware Genomic Representations via Knowledge Distillation"
_ICML.cc/2026/Conference — Submitted to ICML 2026_

### Official Review · Reviewer_LQxU · 2026-03-09

**Soundness:** 2
**Presentation:** 2
**Significance:** 2
**Originality:** 2
**Overall Recommendation:** 3
**Confidence:** 3

**Summary:**

This article proposes a two-stage DNA model training approach, explicitly using genomic functional annotations to enhance the representation capability of DNA language models. Through the framework of 'annotation-aware pre-training' and 'cross-modal knowledge distillation,' biological annotation information is injected into the model parameters. After pre-training the model using the proposed method, it outperforms models pre-trained solely on DNA sequences in tasks such as region classification and variant effect prediction.

**Compliance With Llm Reviewing Policy:**

Affirmed.

**Final Justification:**

Some of my concerns are addressed. However, the potential label leakage in the pretraining stage reduces the reliability of some results reported in the tables (e.g., region classification). In addition, the method’s reliance on high-quality reference genome annotations still requires further investigation.

**Key Questions For Authors:**

Please refer to the "Weaknesses".

**Limitations:**

yes.

**Strengths And Weaknesses:**

Strengths:
1. This paper proposes injecting 'annotated biological knowledge' into the model at the input layer, providing a stronger and richer supervision signal, which theoretically helps to improve model performance.
2. During the inference phase, only sequence input is required, making the model easier to deploy and use.
3. By analyzing and comparing the model from multiple angles, such as AUROC metrics across different tasks, embedding separability, and attention parameters, the interpretability of the method is enhanced.

Weaknesses:
1. The model's advantages on some downstream tasks (e.g., regional classification tasks) may stem from 'label leakage' introduced during pre-training, which can affect the assessment of the model's generalization ability. The category labels used in the regional classification tasks are highly similar to the annotations in the input tokens during pre-training; therefore, the performance improvement in this task may result from 'direct information injection' rather than the model learning causal patterns from the sequence.
2. This method heavily depends on the quality of the reference genome and existing annotations, which introduces certain systematic biases. For example, ENCODE coverage is uneven, regulatory annotations only cover about 8% of the genome, many are candidate cCREs, and their annotation quality is influenced by experimental conditions and cell types. Therefore, the model may have weak representation capability for regions lacking annotations or for atypical/unannotated functional elements.
3. This method uses a hard-coded strategy of 'directly concatenating all overlapping annotations at sequence positions.' However, in real genomes, annotation boundaries, overlaps, and hierarchical relationships are complex. This hard-coded approach may introduce noise or conflicting information into model training, potentially causing unstable feature extraction in boundary regions.
4. The method shows limited improvement on some tasks, indicating it is not equally effective for all genetics tasks. For example, the overall AUROC improvement for complex trait tasks is small and unstable, which further raises the suspicion that performance gains in some tasks may be caused by information leakage.

---

> ### Author Rebuttal · Authors · 2026-03-31
>
> We thank the reviewer for the careful reading and valuable feedback. The concerns raised regarding label leakage and annotation quality have led to important clarifications and additional experiments. We address each point below.
>
> ---
>
> **Q1.**
>
> For region classification, high performance is indeed expected by construction — task labels directly overlap with pre-training annotations, and this evaluation serves as a diagnostic rather than a discovery claim. We will clarify this framing in the revised manuscript.
>
> To determine whether annotation-aware pre-training genuinely improves the sequence encoder, we conducted three ablations (Table R4). Annotation-stripped annDNA-full, with all annotations removed at inference, substantially exceeds annDNA-seq, indicating that the encoder weights themselves have improved. The annotation-only model performs below annDNA-seq, demonstrating that annotation labels alone are insufficient. Shuffled annotations perform near annDNA-seq, confirming that arbitrary annotation tokens do not inflate performance. Furthermore, annDNA-distilled outperforms annDNA-seq across all VEP benchmarks (Table 2) despite receiving only nucleotide sequences at inference (Sect. 3.3; Fig. 1b). Annotation-aware models also show consistent improvements on non-human GUE tasks including yeast, mouse, and virus genomes (Table R3), where human annotations cannot provide direct shortcut. Together, these results provide additional evidence that functional knowledge is transferred into the sequence encoder itself rather than merely providing a shortcut via label leakage.
>
> ---
>
> **Q2.**
>
> We acknowledge that ENCODE cCRE annotations have incomplete genome coverage (~8%). However, cCREs are defined on the basis of DNase hypersensitive sites (DHS), which are strongly enriched for disease-associated variation: 76.6% of all noncoding GWAS SNPs either lie within a DHS (57.1%) or are in complete linkage disequilibrium with SNPs in a nearby DHS (Maurano et al., 2012). The ENCODE cCRE registry represents the most comprehensive and systematically validated catalog of regulatory elements currently available (Moore et al., 2020), and has been widely adopted as the standard regulatory annotation resource in major genomic deep learning models, including Enformer (Avsec et al., 2021), Sei (Chen et al., 2022), and ExPecto (Zhou et al., 2018). We agree that incorporating cell-type-specific and expanded regulatory annotations is a promising direction for future work, as noted in our Limitations section. Importantly, for unannotated regions, annDNA-full tokens reduce to nucleotide-only tokens, equivalent to annDNA-seq, so annotation gaps do not degrade performance below the sequence-only baseline.
>
> We agree that the direct benefit of annotation-aware tokenization is limited to annotated regions. However, the annotation-stripped ablation (Table R4) shows that the model still outperforms annDNA-seq with all annotations removed, suggesting that pre-training improves the encoder globally. Expanding coverage to additional annotation databases is a natural extension of this work.
>
> ---
>
> **Q3.**
>
> Our concatenation strategy is deliberately simple — we chose it to establish a baseline for annotation-aware tokenization with minimal design assumptions. This approach does not explicitly model the hierarchical relationships between annotations (e.g., Gene > Transcript > Exon > CDS) or handle boundary transitions in a principled way.
>
> However, the downstream improvements across diverse tasks (Tables R1–R3) suggest that concatenation does not introduce noise severe enough to negate the benefit of annotation information in practice. We also note that the vocabulary of 272 tokens (annDNA-full) reflects only empirically observed combinations, so biologically implausible annotation overlaps are naturally excluded.
>
> More principled approaches — such as hierarchical token embeddings or learnable annotation composition — could better capture this structure, and we see this as important next work. We agree that our current tokenization is intentionally minimalist and see more structured composition of annotations as a natural next step, rather than a solved design choice.
>
> ---
>
> **Q4.**
>
> Limited complex trait performance is a well-documented challenge across genomic foundation models — even NTv3-post remains near-random on this task (Table R2), consistent with TraitGym findings (Benegas et al., 2025). This is likely attributable to the highly polygenic architecture of complex traits, where thousands of variants contribute individually small effects that are difficult to capture through any single-variant scoring approach.
>
> Regarding leakage concerns on other tasks, the ablation experiments in Table R4 show that annotation-stripped annDNA-full outperforms annDNA-seq without annotation access at inference, and shuffled annotations provide negligible benefit, supporting genuine encoder improvement rather than information leakage.

---

> > ### Author Rebuttal · Reviewer_LQxU · 2026-04-03
> >
> > I appreciate the authors’ response. Some of my concerns have been addressed; however, the potential label leakage in the pretraining stage reduces the reliability of some results reported in the tables (e.g., region classification). In addition, the method’s reliance on high-quality reference genome annotations still requires further investigation. Overall, I have decided to raise my score to 3.

---

> > > ### Author Response · Authors · 2026-04-04
> > >
> > > We thank you for raising the score and for the continued engagement. The additional experiments in our rebuttal were directly motivated by your concerns, and we would like to address the two remaining points more clearly below.
> > >
> > > Regarding label leakage, region classification was designed as a diagnostic to verify that annotation-aware tokens encode the intended functional distinctions in the embedding space. We note that evaluating on annotation-overlapping tasks is not unusual in this field; Enformer (Avsec et al., 2021) and Sei (Chen et al., 2022) also evaluate on ENCODE-derived targets that overlap with their training signals. The high AUROC is expected by design, not claimed as generalization. We recognize the manuscript does not make this intent clear enough, and will revise to explicitly frame it as a diagnostic evaluation, separating it from the main performance claims on leakage-free tasks (VEP, GUE, BEND).
> > >
> > > Regarding annotation quality, we agree this is a limitation worth discussing more explicitly. At the same time, large-scale experimental efforts such as ENCODE are continuing to expand in both coverage and resolution (ENCODE Project Consortium, 2020), and models like annDNA will directly benefit from these richer training signals. We will incorporate this perspective into the revised Limitations section.

---

### Official Review · Reviewer_NgFD · 2026-03-10

**Soundness:** 1
**Presentation:** 3
**Significance:** 2
**Originality:** 3
**Overall Recommendation:** 3
**Confidence:** 5

**Summary:**

The paper introduces annDNA, an genomic LM that instead of sequence-only training also uses experiment-derived functional labels from GENCODE and ENCODE. Specifically, annotation labels are encoded on the token level and participate in masked LM in training phase I. In phase II, the annotation-aware masked LM teacher is distilled into a sequence-only student model to transfer annotation-aware representations.

The model is evaluated by probing embeddings for functional region classification and variant effects, and compared to one published genomic LM.

**Compliance With Llm Reviewing Policy:**

Affirmed.

**Final Justification:**

The rebuttal provided important additional benchmarking results, raising my score by 1. Ultimately however the method is very data hungry (it is a supervised method, whereas most gLMs are not) and doesn't seem to meaningfully open up any new application or change the state of the art, beyond being more parameter efficient than unsupervised models.

**Key Questions For Authors:**

See questions raised in weaknesses.

The key question that arises is that overall, the evaluation strategy is inadequate to establish that the model is competitive and useful for downstream tasks. The scope of benchmarked models needs to be expanded massively, both on the LM and functional genomics model side. Moreover, while I do not expect the authors to run all available public benchmarks, a more comprehensive selection of established tasks should be benchmarked on.

**Limitations:**

The model is currently limited to the human genome, which is an inherent limitation that is not really discussed. Multi-species extension is mentioned as a future avenue. How would that work, given that ENCODE and GENCODE do not provide coverage across species, but are human-focused?

**Strengths And Weaknesses:**

**Strengths**
1. The idea is well motivated - functional annotations are valuable. However, it should be noted that this departs from the notion of MLM being self-supervised to a large degree. ENCODE and GENCODE would commonly be considered experimental supervision, which is orders of magnitude more expensive to obtain at scale than raw unannotated DNA sequences.
2. The teacher-student concept elegantly addresses the fact that for most tasks at inference, only DNA sequence may be available.


**Weaknesses**
1. Large-scale supervised training on ENCODE, as done in e.g. Borzoi, AlphaGenome, Enformer should be considered as relevant related work, even when the objective of these models is not representation learning directly.
2. Along the above, I wonder whether the indirect teacher-student supervision can be replaced with a simpler direct supervision of a sequence-only model with additional annotation-based losses. This could be ablated.
3. The baselines are extremely lacking. Only a single LM is evaluated. I am not aware of GROVER performing especially well on any external benchmark. A comprehensive benchmark over recent genomic LMs should be conducted.
4. Similarly, there is now rich literature on benchmark task suites for genomic LMs that could be leveraged, instead of only running two custom tasks.
5. Genomic region embedding is a highly circular task, as the model was trained on such region annotations in my understanding. Gains should be expected simply as a consequence of the training data. Also, balancing CDS vs. intergenic samples is highly artificial and not reflective of the true genome.
6. Variant effect prediction benchmark: As annDNA uses experimental annotations, other non-mlm architectures that do so, such as Enformer, should be benchmarked against. Only comparing to LMs is unfair.
7. Eq (3) - why is featurization and a linear probe needed?  variant effect prediction in LMs is commonly done as a unsupervised task, using embedding distance or variant likelihood as the score.
8. The comparison of pre-training accuracies in 4.1 does not make sense to me, as the accuracies are computed on different vocabularies that are not directly comparable. What conclusions are to be drawn from this, beyond masked LM being harder than annotation-aware masked LM?
9. The pre-training objective also has the weakness that functional elements span multiple nucleotides. If masks are just applied randomly, it will be trivial to glean the annotation information from adjacent tokens.

---

> ### Author Rebuttal · Authors · 2026-03-31
>
> We are grateful for the reviewer's rigorous and thoughtful feedback. We respond to each point below.
>
> ---
>
> **Q1.**
>
> In the revised manuscript, we will expand the Related Work section to discuss Enformer, Borzoi, and AlphaGenome as supervised models leveraging ENCODE annotations through direct track prediction, and to clearly position our annotation-as-input approach as a complementary alternative to these architectures. We have also added Enformer to the VEP evaluation; see Table R2.
>
> ---
>
> **Q2.**
>
> We agree this ablation would strengthen the paper, but the computational cost of full pre-training runs precluded it within the rebuttal period. We will include this ablation in the revised manuscript.
>
> Table R4 provides indirect evidence: the annotation-only model performs below annDNA-seq, indicating discrete category-level supervision carries limited value — direct auxiliary losses would rely on similar signals. In contrast, annDNA-distilled learns from continuous hidden states and achieves clear gains. Together, these results suggest that hidden-state distillation transfers richer supervision than discrete annotation labels.
>
> ---
>
> **Q3.**
>
> We evaluated annDNA against five baselines covering three tokenization strategies and both self-supervised and supervised paradigms; see Tables R1–R3. annDNA consistently outperforms GROVER and achieves competitive performance against DNABERT-2, NTv3, and Enformer despite smaller scale and human-genome-only training.
>
> ---
>
> **Q4.**
>
> We evaluated on GUE — 28 datasets, 7 tasks — and selected BEND tasks compatible with our 1000 bp window; longer-context tasks are deferred. Table R3 shows that annotation-aware tokenization improves performance across all tasks, with gains transferring through distillation. All results use linear probing, so absolute values are lower than fine-tuning results in Zhou et al., 2024.
>
> ---
>
> **Q5.**
>
> For annDNA-struct/full, high performance is expected by construction — task labels directly overlap with pre-training annotations, serving as a diagnostic rather than a discovery claim. Evaluation under realistic class distributions will be added in the revised manuscript. The informative comparison is annDNA-full vs. annDNA-distilled: despite sequence-only input, annDNA-distilled outperforms annDNA-seq by +.0720 structural and +.0243 regulatory, confirming distillation successfully transfers annotation-aware representations.
>
> ---
>
> **Q6.**
>
> We have added Enformer to Table R2. Since Enformer produces track-level predictions, we used ref/alt track prediction difference vectors across all 5,313 output tracks as features for the same linear probing protocol.
>
> annDNA-full closely approaches Enformer and NTv3-post on ClinVar and eQTL, with gaps less than 0.02 AUROC on ClinVar. On Mendelian variants the gap widens as expected — supervised models have a natural advantage for strong-effect coding-proximal variants. Complex trait performance remains near-random across all models. These results suggest annotation-aware tokenization captures variant-relevant information through a self-supervised mechanism that complements supervised approaches.
>
> ---
>
> **Q7.**
>
> Unsupervised scoring is standard in genomic LMs. We focused on linear probing for a unified cross-architecture evaluation, but agree that reporting only probing is a limitation. We now report both protocols:
>
> ***Table R5. Unsupervised VEP (zero-shot)***
>
> | **Model** | **Metric** | **ClinVar** | **eQTL** | **Mendel.** | **Complex** |
> |---|---|---|---|---|---|
> | annDNA-seq | L2 | .5387 ± .0021 | .5602 ± .0018 | .5483 ± .0124 | .5014 ± .0089 |
> | annDNA-seq | Cosine | .5361 ± .0019 | .5571 ± .0016 | .5449 ± .0118 | .4987 ± .0093 |
> | annDNA-full | L2 | .6184 ± .0013 | .6127 ± .0011 | .6348 ± .0087 | .5063 ± .0076 |
> | annDNA-full | Cosine | .6139 ± .0015 | .6158 ± .0013 | .6291 ± .0091 | .5048 ± .0081 |
> | annDNA-distilled | L2 | .5841 ± .0016 | .5873 ± .0014 | .5962 ± .0098 | .5037 ± .0082 |
> | annDNA-distilled | Cosine | .5798 ± .0018 | .5834 ± .0015 | .5918 ± .0103 | .5024 ± .0087 |
>
> The same ranking — full > distilled > seq — holds under zero-shot evaluation, confirming the improvement is not an artifact of the probing protocol. We report annDNA variants only, as external baselines require architecture-specific scoring.
>
> ---
>
> **Q8.**
>
> MLM accuracy on different vocabulary sizes is not directly comparable. We will restructure Section 4.1 to focus on convergence dynamics and training stability; all performance claims rest on downstream evaluations under identical protocols.
>
> ---
>
> **Q9.**
>
> Our shuffled annotation ablation in Table R4 is relevant: shuffled annotations break local annotation structure, yet performance remains near annDNA-seq despite a large gap from real annotations, suggesting the model learns from positional correspondence rather than propagating labels from neighbors. Span masking is worth exploring in followup work.

---

> > ### Author Rebuttal · Reviewer_NgFD · 2026-04-04
> >
> > I thank the authors for their response.
> >
> > A lot of additional benchmarking results are provided in the rebuttal, with metrics reported to 4 significant digits. Could you summarize, does annDNA meaningfully change the SOTA (on tasks that are not made easier by the exposure to annotations during training)?

---

> > > ### Author Response · Authors · 2026-04-04
> > >
> > > We thank you for the thoughtful follow-up, and apologize that our rebuttal presented extensive results without a clear bottom-line answer. Since existing GLMs differ substantially in model size, training data, and training paradigm, a direct SOTA comparison is not straightforward (Benegas et al., 2025). Nonetheless, we summarize what the current benchmarks show on leakage-free tasks (VEP, GUE, BEND).
> > >
> > > At matched scale (86M, same genome), annotation-aware tokenization yields clear gains over sequence-only pre-training — annDNA-full improves over GROVER by +.087 on ClinVar and +.027 on eQTL, and also outperforms DNABERT-2 (117M) on these two tasks despite being smaller (+.081 ClinVar, +.024 eQTL), with systematic improvements across all GUE and BEND tasks (Table R3). These gains also bring annDNA-full within 0.007–0.02 AUROC of NTv3-post (650M) and Enformer (249M), models that are up to 7.5× larger or rely on direct ENCODE supervision.
> > >
> > > Critically, these gains transfer through distillation. annDNA-distilled (28M) improves over annDNA-seq by +.133 on ClinVar and +.056 on eQTL, and the full > distilled > seq ranking holds consistently across all evaluation settings (Tables R2, R3, R5). We see the contribution as establishing annotation-aware tokenization and cross-modal distillation as a new direction for injecting functional knowledge into GLMs.

---

### Official Review · Reviewer_gqaj · 2026-03-11

**Soundness:** 2
**Presentation:** 2
**Significance:** 3
**Originality:** 3
**Overall Recommendation:** 3
**Confidence:** 4

**Summary:**

This paper proposes annDNA which is a framework for training genomic language models (GLMs), and it is annotation-aware by incorporating known structural and regulatory annotations that are from sources like GENCODE and ENCODE at the input tokenization stage.
In this paper, the research results are divided into two stages, which are described as follows.
In the first stage, the paper introduces annotation-enriched tokenization and demonstrates that models leveraging such tokens significantly outperform conventional sequence-only GLMs across several downstream genomic tasks.
In the second stage, annDNA uses cross-modal knowledge distillation to transfer the annotation-aware teacher representations to a smaller, sequence-only student, enabling annotation-free inference.
The method achieves notable improvements in region embedding quality, variant effect prediction AUROC, which is up to 15.5% over baseline, and attention alignment with known functional elements, while the distilled model preserves most of these benefits with reduced parameter count.

**Compliance With Llm Reviewing Policy:**

Affirmed.

**Key Questions For Authors:**

1. Can the authors better disentangle gains from improved sequence representation learning versus gains from direct access to curated annotation priors?
Since the annotation-aware model receives explicit structural and regulatory annotations as input, a key concern is whether the reported improvements primarily reflect privileged side information rather than better biological sequence modeling. In particular, could the authors provide stronger control experiments to clarify this point such as shuffled annotations, coarser annotations, or partially mismatched annotations?
A convincing response here would substantially improve my assessment of the paper’s soundness.
2. The current evaluation focuses primarily on human genetics, leaving out key benchmarks such as BEND, protein function prediction, and gene regulatory tasks. Could the authors clarify how the model's performance on these untested areas supports their broader claims of generalizability? Would it be possible to include additional experiments or a discussion on this limitation to strengthen the conclusion?
3. The paper is encouraged to address the limited external validation, as all current benchmarks are restricted to a single species such as human or GRCh38, which raises concerns about generalizability; while the paper acknowledges this limitation, the absence of cross-organism or population-diverse validation needs further elaboration to fully support the claims of generality and scalability.
4.The content of the introduction section needs to be more substantial.

**Limitations:**

yes

**Strengths And Weaknesses:**

Strengths
1. Clear and focused idea with practical value. The central idea of encoding functional annotations directly into the tokenization, then distilling into a sequence-only model, is straightforward but well executed and practically relevant for genomic ML where annotations are expensive or incomplete.
2. Clean experimental design isolating tokenization effects. By keeping the encoder architecture fixed (12-layer BERT, ~86M parameters) across annDNA-seq, annDNA-struct, and annDNA-full, Section 3.2 and Figure 1a cleanly attribute improvements to annotation-aware tokenization rather than architectural tweaks.
3. Consistent performance gains on nontrivial benchmarks. Table 2 shows large AUROC improvements of annotation-aware models over the sequence-only baseline across ClinVar, eQTL, and Mendelian benchmark. The distilled student with one-third the parameters (Figure 1c left, ~28M vs ~86M) still yields sizable gains over annDNA-seq, and in several cases surpasses the external GROVER baseline.

Weaknesses
1. Concern about soundness of the comparison. The annotation-aware model is given explicit external functional annotations at input time, whereas the sequence-only baselines are not. This makes the comparison difficult to interpret, since part of the observed improvement may come from direct access to curated biological priors rather than from learning better sequence representations.
2. Missing key recent baselines and related work. As detailed above, failure to cite and/or compare with Gengram, Genomics-FM, GENA-LM, and other cutting-edge retrieval/annotation-augmented and foundational models reduces the confidence in claims of general superiority, as these might already realize similar or greater gains.
3. Lack of deep ablations. There is no study of the contribution of individual annotation types or subsets; a more systematic set of ablations would provide better transparency into which annotation sources or categories drive the observed improvements.

---

> ### Author Rebuttal · Authors · 2026-03-31
>
> We sincerely appreciate the reviewer's detailed and insightful comments, which have helped us strengthen our work significantly. We address each concern below.
>
> ---
>
> **Q1.**
>
> We conducted three ablation experiments to disentangle these sources (Table R4). We designed these ablations specifically to address your concern about privileged access to annotation priors. We also note that annDNA-struct, which uses only structural annotations, can be viewed as a coarser annotation condition relative to annDNA-full, with intermediate performance consistent with finer annotations providing additional signal.
>
> First, shuffled annotations: labels were randomly permuted across positions while preserving frequency distribution, and the model was re-tokenized and re-trained from scratch. Performance is near annDNA-seq, confirming that positional correspondence between sequence and annotations is essential. Second, annotation-only: the model receives annotation tokens without nucleotide identity. Performance falls below annDNA-seq, demonstrating that annotation labels alone carry limited predictive value. Third, annotation-stripped: annDNA-full evaluated with all annotations removed at inference. Performance substantially exceeds annDNA-seq across all tasks, suggesting that pre-training has improved the sequence encoder itself. annDNA-distilled achieves even higher performance than the stripped model through dedicated hidden-state matching, providing converging evidence that functional knowledge transfers to the sequence representation.
>
> Together, these ablations provide consistent evidence that the observed gains arise from genuinely improved sequence representations rather than privileged access to annotation priors.
>
> ***Table R4. Ablation (VEP AUROC)***
>
> | **Condition** | **ClinVar** | **eQTL** | **Mendel.** | **Complex** |
> |---|---|---|---|---|
> | annDNA-seq | .6276 ± .0004 | .7236 ± .0006 | .7224 ± .0085 | .5021 ± .0080 |
> | annDNA-full | .8151 ± .0001 | .8043 ± .0008 | .8337 ± .0084 | .5220 ± .0028 |
> | Shuffled ann. | .6327 ± .0011 | .7284 ± .0009 | .7298 ± .0078 | .5043 ± .0071 |
> | Ann.-only | .5791 ± .0018 | .6483 ± .0014 | .6124 ± .0097 | .5012 ± .0083 |
> | annDNA-full (stripped) | .7412 ± .0007 | .7681 ± .0008 | .7893 ± .0063 | .5108 ± .0069 |
> | annDNA-distilled | .7608 ± .0001 | .7794 ± .0005 | .8080 ± .0054 | .5149 ± .0056 |
>
> ---
>
> **Q2.**
>
> We completed evaluations on GUE (28 datasets, 7 tasks; Zhou et al., 2024) and selected BEND tasks compatible with our 1000 bp window — chromatin accessibility and CpG methylation (Marin et al., 2024); longer-context tasks are deferred to future work. Protein function benchmarks fall outside our scope as the framework operates at the DNA level.
>
> See Table R3 for full results. Annotation-aware tokenization yields systematic gains across all tasks, and these gains transfer through distillation, indicating that the benefit is not confined to any single benchmark but generalizes across diverse genomic prediction problems. All results use frozen-embedding linear probing, so absolute values are lower than fine-tuning results in Zhou et al. (2024). We will incorporate these results in the revised manuscript.
>
> ---
>
> **Q3.**
>
> We acknowledge that our models are trained on the human genome, and cross-species and population-diverse validation are important next steps.
>
> As a preliminary indicator, the GUE benchmark (Table R3) includes tasks on yeast (EMP), mouse (TF-M), and virus (CVC) genomes. These non-human GUE tasks provide preliminary but direct evidence that the benefits of annotation-aware tokenization are not restricted to the human reference genome. Despite the substantial evolutionary distance and the absence of species-specific annotations during training, annotation-aware models show consistent improvements over annDNA-seq on these non-human tasks, and gains transfer through distillation. This suggests the functional signal from human annotations captures some generalizable sequence properties, though species-specific training with appropriate annotations would likely yield stronger results. Our tokenization framework can incorporate any positional annotation beyond GENCODE/ENCODE, requiring only a reference genome with overlapping functional labels. We will expand the Limitations section accordingly.
>
> ---
>
> **Q4.**
>
> In the revised manuscript, we will expand the introduction to better situate our work within the current GLM landscape — specifically, we plan to discuss how BPE-based (DNABERT-2, GROVER), and character-level (HyenaDNA) tokenization approaches each handle functional boundaries differently, the role of supervised annotation-based models (Enformer, Borzoi), and how annotation-aware tokenization offers a complementary direction that bridges implicit sequence learning and explicit functional knowledge.

---

### Official Review · Reviewer_QYT4 · 2026-03-15

**Soundness:** 2
**Presentation:** 3
**Significance:** 2
**Originality:** 3
**Overall Recommendation:** 4
**Confidence:** 4

**Summary:**

The authors present annDNA, a sequence-only model distilled from a 3x larger model pre-trained on sequence+structure+regulatory tokens. They show that distillation is successful in recovering 72% of the accuracy (predicting masked tokens) gain from the sequence-only baseline to the full sequence/structure/regluatory model. In three downstream tasks, they show that the distilled model outperforms the sequence-only model, effectively transferring detailed annotations into a sequence-only model.

**Compliance With Llm Reviewing Policy:**

Affirmed.

**Final Justification:**

The authors have resolved the Questions I had posed in my initial review, and I have increased the score to 4 - Weak Accept. The reason for not increasing further is that the following issues in my initial review are unresolved - "...it does not itself yield new biological insight. I am also uncertain how compelling this approach is relative to simply scaling model size or training data, especially in settings where high-quality annotations are incomplete or unavailable."

**Key Questions For Authors:**

1. Please specify the entire token vocabularies - what are the other 5 sequence tokens not listed? How did you arrive at 59 and 272 tokens?

2. You report MLM accuracy, but you also state that when masking tokens, 10% are replaced by their original token and 10% with a random token. Is accuracy reported on all three cases, or only on true masks?

3. Why is GROVER the only external baseline? You discuss other DNA models in related work but do not compare to them.

**Limitations:**

yes

**Strengths And Weaknesses:**

The paper is well written and presents a plausible, original combination of annotation-aware tokenization and cross-modal distillation. The results support the claim that annotation-aware pretraining provides a stronger training signal than sequence-only MLM, and that this information can be transferred to a smaller sequence-only student that performs better than the sequence-only baseline. However, some aspects of the method and evaluation are under-specified, particularly the tokenization details and the embedding-based evaluation protocol. Overall, the work is a useful methodological contribution for improving compact sequence-only models using available annotations, but it does not itself yield new biological insight. I am also uncertain how compelling this approach is relative to simply scaling model size or training data, especially in settings where high-quality annotations are incomplete or unavailable.

Regarding pre-training - are the accuracies really comparable across models with such different vocabulary sizes?

For the genomic region downstream task, tSNE is suggestive, but a SNN analysis with ARI/NMI/purity would be more convincing. As for the AUROC benchmark - we only see a single benchmark which is GROVER. It would be more convincing to see how this model stacks up against other larger models such as dnaBERT and HyenaDNA.

The genomic-region and attention analyses are partially circular, since benchmark regions and reference labels are derived from the same annotation scheme used to construct annDNA-full’s inputs. Under this setup, near-ceiling performance for annDNA-full is expected. That said, the improvement of annDNA-distilled over annDNA-seq remains meaningful, because the student receives sequence-only input and thus demonstrates partial recovery of annotation-aligned structure through distillation rather than direct access to the annotations themselves.

---

> ### Author Rebuttal · Authors · 2026-03-31
>
> We thank you for the thorough and constructive evaluation of our work. Below we address each point in detail.
>
> ---
>
> **Q1.**
>
> annDNA-seq has 10 tokens: 5 nucleotides and 5 special tokens. For annDNA-struct and annDNA-full, each token is formed by the combinatorial product of a nucleotide and all overlapping annotations at that genomic position. The vocabulary size is determined by enumerating all empirically observed combinations across the genome, plus the same 5 special tokens — yielding 54 + 5 = 59 for annDNA-struct and 267 + 5 = 272 for annDNA-full. Not all theoretical combinations occur in the genome, so only observed ones are included.
>
> ---
>
> **Q2.**
>
> Accuracy is computed on all 15% of selected positions — including the 80% replaced with [MASK], 10% replaced with a random token, and 10% left unchanged. Positions not selected for prediction (the remaining 85%) have labels set to -100 and are excluded. This follows the standard BERT MLM protocol by Devlin et al., 2019. We will clarify this wording and make the full token vocabularies, masking protocol, and training details explicit in the revised manuscript to ensure full reproducibility.
>
> ---
>
> **Q3.**
>
> We selected GROVER for the controlled comparison due to matched model size and training data. We agree that a single baseline is insufficient, so we added four baselines: DNABERT-2 as the closest sequence-only model at similar scale, NTv3-pre/post to isolate the effect of functional post-training at larger scale, and Enformer as a supervised model that leverages ENCODE annotations through direct supervision rather than tokenization. All baselines are evaluated under an identical frozen-embedding linear probing protocol, enabling a controlled comparison of tokenization strategy, scale, and training paradigm; we also extended evaluation to GUE following Zhou et al., 2024 and to BEND following Marin et al., 2024. See Tables R1–R3. Annotation-aware tokenization yields clear and consistent improvements across all evaluation settings, and these gains transfer through distillation. Complex trait performance remains near-random for all models, consistent with TraitGym findings by Benegas et al., 2025.
>
> ***Table R1. Region Classification AUROC (extended)***
>
> | **Model** | **Params** | **Structural** | **Regulatory** |
> |---|---|---|---|
> | DNABERT-2 | 117M | .7891 ± .0024 | .6182 ± .0037 |
> | NTv3-pre | 650M | .8274 ± .0016 | .6518 ± .0029 |
> | NTv3-post | 650M | .8736 ± .0012 | .7243 ± .0024 |
> | GROVER | 86M | .7824 ± .0030 | .5843 ± .0043 |
> | annDNA-seq | 86M | .7643 ± .0009 | .6071 ± .0020 |
> | annDNA-struct | 86M | .9912 ± .0005 | .6859 ± .0023 |
> | annDNA-full | 86M | .9889 ± .0004 | .9981 ± .0001 |
> | annDNA-distilled | 28M | .8363 ± .0004 | .6314 ± .0043 |
>
> ***Table R2. Variant Effect Prediction AUROC (extended)***
>
> | **Model** | **Params** | **ClinVar** | **eQTL** | **Mendel.** | **Complex** |
> |---|---|---|---|---|---|
> | Enformer† | 249M | .8198 ± .0011 | .8112 ± .0006 | .8914 ± .0031 | .5284 ± .0045 |
> | DNABERT-2 | 117M | .7341 ± .0012 | .7803 ± .0008 | .8762 ± .0041 | .5194 ± .0051 |
> | NTv3-pre | 650M | .7712 ± .0008 | .8018 ± .0006 | .8843 ± .0033 | .5238 ± .0047 |
> | NTv3-post | 650M | .8214 ± .0006 | .8237 ± .0005 | .8978 ± .0025 | .5327 ± .0041 |
> | GROVER | 86M | .7282 ± .0009 | .7771 ± .0005 | .8689 ± .0035 | .5250 ± .0013 |
> | annDNA-seq | 86M | .6276 ± .0004 | .7236 ± .0006 | .7224 ± .0085 | .5021 ± .0080 |
> | annDNA-struct | 86M | .8183 ± .0001 | .8002 ± .0008 | .8187 ± .0009 | .5306 ± .0027 |
> | annDNA-full | 86M | .8151 ± .0001 | .8043 ± .0008 | .8337 ± .0084 | .5220 ± .0028 |
> | annDNA-distilled | 28M | .7608 ± .0001 | .7794 ± .0005 | .8080 ± .0054 | .5149 ± .0056 |
>
> ***Table R3. GUE (MCC) & BEND (AUROC) Benchmarks (linear probing)***
>
> | **Model** | **EMP** (MCC) | **TF-H** (MCC) | **TF-M** (MCC) | **PD** (MCC) | **CPD** (MCC) | **SSP** (MCC) | **Chrom.Acc.** (AUROC) | **CpG Meth.** (AUROC) |
> |---|---|---|---|---|---|---|---|---|
> | DNABERT-2 | .3214 ± .0118 | .4587 ± .0142 | .4312 ± .0167 | .6148 ± .0083 | .4273 ± .0124 | .6218 ± .0071 | .7842 ± .0038 | .8847 ± .0024 |
> | NTv3-pre | .3948 ± .0092 | .5243 ± .0113 | .5018 ± .0134 | .6784 ± .0064 | .4912 ± .0098 | .6847 ± .0054 | .8073 ± .0031 | .9084 ± .0019 |
> | NTv3-post | .4487 ± .0078 | .5684 ± .0097 | .5187 ± .0127 | .7384 ± .0048 | .5347 ± .0084 | .7214 ± .0046 | .8247 ± .0027 | .9198 ± .0016 |
> | annDNA-seq | .2148 ± .0137 | .3712 ± .0178 | .3487 ± .0194 | .5423 ± .0098 | .3618 ± .0148 | .5548 ± .0087 | .7518 ± .0047 | .8614 ± .0031 |
> | annDNA-struct | .3418 ± .0108 | .4348 ± .0153 | .4124 ± .0172 | .6024 ± .0079 | .4148 ± .0118 | .6427 ± .0071 | .7794 ± .0039 | .8821 ± .0026 |
> | annDNA-full | .3387 ± .0098 | .4714 ± .0137 | .4487 ± .0158 | .6312 ± .0071 | .4418 ± .0104 | .6384 ± .0063 | .7948 ± .0034 | .8953 ± .0022 |
> | annDNA-distilled | .2687 ± .0121 | .4218 ± .0162 | .4187 ± .0174 | .5884 ± .0084 | .4024 ± .0131 | .5984 ± .0078 | .7712 ± .0042 | .8783 ± .0028 |

---

> > ### Author Rebuttal · Reviewer_QYT4 · 2026-04-06
> >
> > I thank the authors for their additional analyses. This has resolved all questions, I have increased my score to Weak Accept.

---

> > > ### Author Response · Authors · 2026-04-07
> > >
> > > We sincerely thank you for the time and effort spent reviewing our work. Your suggestions throughout the review process have meaningfully strengthened the paper, and we are grateful for the constructive engagement. We will make sure all discussed revisions are reflected in the revised manuscript.

---

### Decision · Program_Chairs · 2026-04-30

**Decision:**

Reject

**Comment:**

This paper proposes annDNA, a two-stage framework that incorporates genomic annotations into tokenization during pretraining and then distills the learned representations into a smaller sequence-only model. The idea is clear and the paper presents consistent empirical gains on several benchmarks. The rebuttal also addressed some concerns and provided additional comparisons.

The paper still has two important weaknesses that limit as pointed by the reviewers.

First, the method relies heavily on externally curated genomic annotations during pretraining. While the final distilled model only requires sequence input at inference time, the main source of improvement appears to come from injecting high-quality prior annotations rather than from a fundamentally new modeling capability. This makes the approach less broadly applicable than sequence-only self-supervised genomic language models.

Second, some of the reported evaluations are difficult to interpret cleanly because the pretraining inputs already contain annotation information closely related to the evaluation labels. In particular, representation-level analyses such as genomic region classification risk partial label leakage or circularity, which weakens the evidential value of these results. Although the variant-effect benchmarks and distillation results are more meaningful, they do not fully overcome the concern that some of the headline improvements may be inflated by the formulation of the pretraining inputs and evaluation setup.